# Federated Learning with Online Adaptive Heterogeneous Local Models

**Hanhan Zhou**
The George Washington University
Washington, DC
hanhan@gwu.edu

**Tian Lan**
The George Washington University
Washington, DC
tlan@gwu.edu

**Guru Venkataramani**
The George Washington University
Washington, DC
guruv@gwu.edu

**Wenbo Ding**
Tsinghua-Berkeley Shenzhen Institute
Shenzhen, China
ding.wenbo@sz.tsinghua.edu.cn

## Abstract

In Federated Learning, one of the biggest challenges is that client devices often have drastically different computation and communication resources for local updates. To this end, recent research efforts have focused on training heterogeneous local models that are obtained by adaptively pruning a shared global model. Despite the empirical success, theoretical analysis of the convergence of these heterogeneous FL algorithms remains an open question. In this paper, we establish sufficient conditions for any FL algorithms with heterogeneous local models to converge to a neighborhood of a stationary point of standard FL at a rate of $O(\frac{1}{\sqrt{Q}})$. For general smooth cost functions and under standard assumptions, our analysis illuminates two key factors impacting the optimality gap between heterogeneous and standard FL: pruning-induced noise and minimum coverage index, advocating a joint design strategy of local models' pruning masks in heterogeneous FL algorithms. The results are numerically validated on MNIST and CIFAR-10 datasets.

## 1 Introduction

Federated Learning (FL) [1] allows distributed clients to collaborate and train a centralized global model without the transmission of local data. In practice, mobile and edge devices that are equipped with drastically different computation and communication capabilities are becoming the dominant source for FL [2]. This has prompted significant recent attention to a family of FL algorithms relying on training heterogeneous local models (often obtained through pruning a shared global model) for global aggregation. It includes algorithms such as HeteroFL [3] that employ fixed heterogeneous local models, algorithms utilizing pre-trained pruning masks following "The Lottery Ticket Hypothesis" [4], as well as algorithms like PruneFL [5] that adaptively prune local models during training. However, the success of these algorithms has only been demonstrated empirically (e.g., [2, 5, 3]). Unlike standard FL that has received rigorous analysis [6–9], the convergence of heterogeneous FL algorithms is still an open question.

This paper tackles the following questions – Given a heterogeneous FL algorithm that trains a shared global model through a sequence of time-varying and client-dependent local models, what conditions can guarantee its convergence? How do the trained models compare to that of standard FL? There have been many existing efforts in establishing convergence guarantees for FL algorithms, such as

Workshop on Federated Learning: Recent Advances and New Challenges, in Conjunction with NeurIPS 2022 (FL-NeurIPS'22). This workshop does not have official proceedings and this paper is non-archival.

the popular FedAvg [1], on both IID [1] and non-IIDdata distributions, but all rely on the assumption that local models share the same uniform structure as the global model. Training heterogeneous local models (which could adaptively change both over time and across clients) in FL is desirable due to its ability to adapt to resource constraints and training outcomes.

For general smooth cost functions under standard FL assumptions, we prove that heterogeneous FL algorithms satisfying certain sufficient conditions can indeed converge to a neighborhood of a stationary point of standard FL (with a small optimality gap that is characterized in our analysis), at a rate of $O(\frac{1}{\sqrt{Q}})$ in $Q$ communication rounds. We prove a new upperbound and show that the optimality gap (between heterogeneous and standard FL) is affected by both pruning-induced noise (as identified in single-model pruning) and a new notion of minimum coverage index in FL (i.e., any parameters in the global model are included in at least $\Gamma_{\min}$ local models). Our results also motivate a joint design of efficient local-model pruning strategies (e.g., leveraging [10–12]) for heterogeneous FL to have comparable convergence with standard FL. It captures a number of existing FL algorithms and provides a general convergence guarantee. We perform extensive experiments on MNIST and CIFAR10 datasets. Our numerical evaluations validate the sufficient conditions established in our convergence analysis.

## 2   Related Work

**Federated Averaging and Communication Efficient FL**. FedAvg [1] is considered the first and the most commonly used federated learning algorithm . Several works have shown the convergence of FedAvg under several different settings with both homogeneous (IID) data [6, 13] and heterogeneous (non-IID) data [9, 7, 8] even with partial clients participation. Specifically, [8] demonstrated LocalSGD achieves $O(\frac{1}{\sqrt{NQ}})$ convergence for non-convex optimization and [9] established a convergence rate of $O(\frac{1}{Q})$ for strongly convex problems on FedAvg, where Q is the number of SGDs and $N$ is the number of participated clients. Several works [14–17] are proposed to further reduce the communication costs. One direction is to use data compression such as quantization [18, 7, 19, 20], sketching [21, 22], split learning [23] and learning with gradient sparsity [24]. None of these work considers the convergence of FL with heterogeneous local models.

**Neural Network Pruning and Sparsification**. To reduce the computation costs of a neural network, neural network pruning is a popular research topic. A magnitude-based prune-from-dense methodology [25–29] is widely used where weights smaller than the certain preset threshold are removed from the network. In addition, there are one-shot pruning initialization [30], iterative pruning approach [31, 32] and adaptive pruning approach [33, 34] that allows network to grow and prune. In [4, 35] a "lottery ticket hypothesis" was proposed that with an optimal substructure of the neural network acquired by weights pruning, directly training a pruned model could reach similar results as pruning a pre-trained network. The other direction is through sparse mask exploration [36–38], where a sparsity in neural networks is maintained during the training process. It is empirically observed [4, 37] training of models with static sparse parameters will converge to a solution with higher loss than models with dynamic sparse training. However, when adaptive model pruning is employed to generate local models in heterogeneous FL, the convergence is an open problem.

**Efficient FL with Heterogeneous Neural Networks**.  Several works are proposed to address the reduction of both computation and communication costs, including one way to utilize lossy compression and dropout techniques[39, 40]. Although early works mainly assume that all local models are to share the same architecture as the global model [41], recent works have empirically demonstrated that federated learning with a heterogeneous client model to save both computation and communication is feasible. PruneFL[5] proposed an approach with adaptive parameter pruning during FL. [42] proposed FL with a personalized and structured sparse mask. HetroFL[3] proposed to generate heterogeneous local models as a subnet of the global network by picking the leading continuous parameters layer-wise with the help of proposed static batch normalization, while [43] finds the small sub-network by applying the structured pruning. Despite their empirical success, they lack theoretical convergence guarantees even in convex optimization settings.

---

[1]Throughout this paper, "IID data" means that the data among local clients are not independent and identically distributed.

# 3 Methodology

## 3.1 Problem Formulation for FL with Heterogeneous Local models

Given an FL algorithm that trains heterogeneous local models for global aggregation, our goal is to analyze its convergence with respect to a stationary point of standard FL. We consider a general formulation where the heterogeneous local models can be obtained using any adaptive model pruning strategies that are both (i) time-varying to enable online adjustment of pruned local models during the entire training process and (ii) different across FL clients with respect to their individual heterogeneous computing resource and network conditions. More formally, we denote the sequence of local models used by a heterogeneous FL algorithm by masks $m_{q,n} \in \{0,1\}^{|\theta|}$, which can vary at any round $q$ and for any client $n$. Let $\theta_q$ denote the global model at the beginning of round $q$ and $\odot$ be the element-wise product. Thus, $\theta_q \odot m_{q,n}$ defines the trainable parameters of the pruned local model[2] for client $n$ in round $q$. Our goal is to find sufficient conditions on such masks $m_{q,n}$ $\forall q, n$ for the convergence of heterogeneous FL. Here, we describe one around (say the $q$th) of the heterogeneous FL algorithm. First, the central server employs a given pruning strategy $\mathbb{P}(\cdot)$ to prune the latest global model $\theta_q$ and broadcast the resulting local models to clients:

$$\theta_{q,n,0} = \theta_q \cdot m_{q,n}, \text{ with } m_{q,n} = \mathbb{P}(\theta_q, n, q), \ \forall n. \tag{1}$$

We note that the pruning strategy $\mathbb{P}(\theta_q, n, q)$ can vary over time $q$ and across clients $n$ in heterogeneous FL. Each client $n$ then trains the pruned local model by performing $T$ local updates (in $T$ epochs):

$$\theta_{q,n,t} = \theta_{q,n,t-1} - \gamma \nabla F_n(\theta_{q,n,t-1}, \xi_{n,t-1}) \odot m_{q,n}, \text{ for } t = 1 \dots, T, \tag{2}$$

where $\gamma$ is the learning rate and $\xi_{n,t-1}$ are independent samples uniformly drawn from local data $D_n$ at client $n$. We note that $\nabla F_n(\theta_{q,n,t-1}, \xi_{n,t-1}) \odot m_{q,n}$ is a local stochastic gradient evaluated using only local parameters in $\theta_{q,n,t-1}$ (available to the heterogeneous local model) and that only locally trainable parameters are updated by the stochastic gradient (via an element-wise product with $m_{q,n}$). Finally, the central server aggregates the local models $\theta_{n,q,T}$ $\forall n$ and produces an updated global model $\theta_{q+1}$. Due to the use of heterogeneous local models, each global parameter is included in a (potentially) different subset of the local models. Let $\mathcal{N}_q^{(i)}$ be the set of clients, whose local models contain the $i$th modeling parameter in round $q$. That is $n \in \mathcal{N}_q^{(i)}$ if $m_{q,n}^{(i)} = \mathbf{1}$ and $n \notin \mathcal{N}_q^{(i)}$ if $m_{q,n}^{(i)} = \mathbf{0}$. Global update of the $i$th parameter is performed by aggregating local models with the parameter available, i.e.,

$$\theta_{q+1}^{(i)} = \frac{1}{|\mathcal{N}_q^{(i)}|} \sum_{n \in \mathcal{N}_q^{(i)}} \theta_{q,n,T}^{(i)}, \ \forall i, \tag{3}$$

where $|\mathcal{N}_q^{(i)}|$ is the number of local models containing the $i$th parameter. We summarize the algorithm details in Algorithm 1 in the Appendix.

## 3.2 Notations and Assumptions

We make the following assumptions that are routinely employed in FL convergence analysis. In particular, Assumptions 1 is a standard and common setting. Assumption 2 follows from [34] (which is for a single-model case) and implies the noise introduced by pruning is relatively small and bounded. Assumptions 3 and 4 are standard for FL convergence analysis following from [45, 46, 8, 9] and assume the stochastic gradients to be bounded and unbiased.

**Assumption 1.** *(Smoothness). Cost functions $F_1, \dots, F_N$ are all L-smooth: $\forall \theta, \phi \in \mathcal{R}^d$ and any $n$, we assume that there exists $L > 0$:*

$$\|\nabla F_n(\theta) - \nabla F_n(\phi)\| \le L\|\theta - \phi\|. \tag{4}$$

**Assumption 2.** *(Pruning-induced Noise). We assume that for some $\delta^2 \in [0,1)$ and any $q, n$, the pruning-induced error is bounded by*

$$\|\theta_q - \theta_q \odot m_{q,n}\|^2 \le \delta^2 \|\theta_q\|^2. \tag{5}$$

---

[2]While a pruned local model has a smaller number of parameters than the global model. We adopt the notations in [5, 44, 34] and use $\theta_q \odot m_{q,n}$ with an element-wise product to denote the pruned local model - only parameter corresponding to a 1-value in the mask is accessible and trainable in the local model.

**Assumption 3.** *(Bounded Gradient). The expected squared norm of stochastic gradients is bounded uniformly, i.e., for constant $G > 0$ and any $n, q, t$:*

$$E \|\nabla F_n(\theta_{q,n,t}, \xi_{q,n,t})\|^2 \leq G. \tag{6}$$

**Assumption 4.** *(Gradient Noise for IID data). Under IID data distribution, for any $q, n, t$, we assume that*

$$\mathbb{E}[\nabla F_n(\theta_{q,n,t}, \xi_{n,t})] = \nabla F(\theta_{q,n,t}), \quad \mathbb{E}\|\nabla F_n(\theta_{q,n,t}, \xi_{n,t}) - \nabla F(\theta_{q,n,t})\|^2 \leq \sigma^2 \tag{7}$$

*for constant $\sigma^2 > 0$ and independent samples $\xi_{n,t}$.*

### 3.3 Convergence Analysis

We now analyze the convergence of heterogeneous FL for general smooth cost functions. We begin with introducing a new notion of minimum covering index, defined in this paper by

$$\Gamma_{\min} = \min_{q,i} |\mathcal{N}_q^{(i)}|, \tag{8}$$

Since $|\mathcal{N}_q^{(i)}|$ is the number of heterogeneous local models containing the $i$th parameter, $\Gamma_{\min}$ measures the minimum occurrence of the parameter in the local models in all rounds. Intuitively, if a parameter is never included in any local models, it is impossible for it to be updated. Thus conditions based on the covering index would be necessary for the convergence toward standard FL. All proofs are collected in the Appendix.

**Theorem 1.** *Under Assumptions 1-4 and for arbitrary masks satisfying $\Gamma_{\min} \geq 1$, heterogeneous FL converges to a small neighborhood of a stationary point of standard FL as follows:*

$$\frac{1}{Q}\sum_{q=1}^{Q}\mathbb{E}\|\nabla F(\theta_q)\|^2 \leq \frac{G_0}{\sqrt{TQ}} + \frac{V_0}{Q} + \frac{I_0}{\Gamma_{\min}} \cdot \frac{\delta^2}{Q}\sum_{q=1}^{Q}\mathbb{E}\|\theta_q\|^2$$

*where $V_0 = 3L^2NG/\Gamma_{\min}$, $I_0 = 3L^2N$, and $G_0 = 4\mathbb{E}[F(\theta_0)] + 6LN\sigma^2/\Gamma_{\min}^2$, are constants depending on the initial model parameters and the gradient noise.*

**Remark 1.** Theorem 1 shows the convergence of heterogenous FL to a neighborhood of a stationary point of standard FL (albeit a small optimality gap due to pruning-induced noise) as long as $\Gamma_{\min} \geq 1$. The result is a bit surprising since $\Gamma_{\min} \geq 1$ only requires each parameter to be included in at least one local model – which is obviously necessary for all parameters to be updated during training. But we show that this is also a sufficient condition for convergence. Moreover, we also establish a convergence rate of $O(\frac{1}{\sqrt{Q}})$ for arbitrary pruning strategies satisfying the condition.

**Remark 2.** Impact of pruning-induced noise. In Assumption 2, we assume the pruning-induced noise is relatively small and bounded with respect to the global model: $\|\theta_q - \theta_q \odot m_{q,n}\|^2 \leq \delta^2 \|\theta_q\|^2$. This is satisfied in practice since most pruning strategies tend to focus on eliminating weights/neurons that are insignificant, therefore keeping $\delta^2$ indeed small. We note that similar observations are made on the convergence of single-model adaptive pruning [33, 34], but the analysis does not extend to FL problems where the fundamental challenge comes from local updates causing heterogeneous local models to diverge before

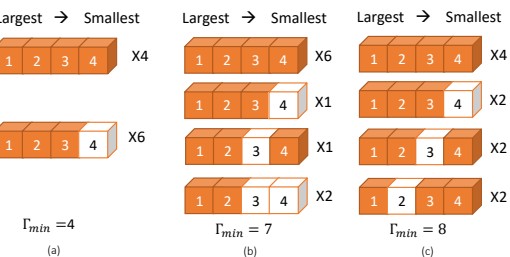

Figure 1: Illustration of three alternative pruning strategies in heterogeneous FL. While all three ensure convergence and have the same total model size, their optimality gap can vary significantly.

the next global aggregation. We note that for heterogeneous FL, pruning will incur an optimality gap $\delta^2 \frac{1}{Q}\sum_{q=1}^{Q}\mathbb{E}\|\theta_q\|^2$ in our convergence analysis, which is proportional to $\delta^2$ and the average model norm (averaged over $Q$). It implies that more aggressive pruning in heterogeneous FL may lead to a larger error, deviating from standard FL at a speed quantified by $\delta^2$. We note that this error is affected by both $\delta^2$ and $\Gamma_{\min}$.

**Remark 3.** Impact of minimum covering index $\Gamma_{\min}$. It turns out that the minimum number of occurrences of any parameter in the local models is a key factor deciding convergence in heterogeneous FL. As $\Gamma_{\min}$ increases, both constants $G_0, V_0$ and the optimality gap decrease. This result is a bit counter-intuitive since certain parameters should be small enough to ignore in pruning. However, recall that our analysis shows the convergence of all parameters in $\theta_q$ with respect to a stationary point of standard FL (rather than for a subset of parameters or to a random point). The more times a parameter is covered by local models, the sooner it gets updated and convergences to the desired target. This is quantified in our analysis by showing that the optimality gap due to pruning noise decreases at the rate of $\Gamma_{\min}$.

**Remark 4.** When the cost function is strongly convex (e.g., for softmax classifier, logistic regression and linear regression with $l_2$-normalization), a stationary point becomes the global optimum. Thus, Theorem 1 shows convergence to a small neighborhood of the global optimum of standard FL for strongly convex cost functions.

**Remark 5.** Theorem 1 also inspires new design criteria for designing adaptive model pruning strategies in heterogeneous FL. Since the optimality gap is affected by both pruning-induced noise $\delta^2$ and minimum covering index $\Gamma_{\min}$, we may prefer strategies with small $\delta^2$ and large $\Gamma_{\min}$, in order to minimize the optimality gap to standard FL. Using these insights, We present numerical examples with different pruning strategy designs in Section 4. Optimal pruning mask design with respect to clients' resource constraints will be considered in future work.

## 4 Empirical Results and Discussion

(a) FL with Different Pruning Techniques on MNIST

| Model | FLOPs | Space | Ratio | $\Gamma_{min}$ | Accuracy IID | Non-IID Local | Non-IID Global |
|-------|-------|-------|-------|------|-----|-------|--------|
| *FullNets* | 158.8K | 1.27M | 1.00 | 10 | 98.01 | 93.82 | 93.59 |
| *WP-L1* | 143.12K | 1.15M | 0.90 | 6 | 98.18 | 95.49 | 95.15 |
| *NP-L1* | 142.9K | 1.14M | 0.90 | 6 | 97.97 | 93.82 | 93.6 |
| *FS-L1* | 142.9K | 1.14M | 0.90 | 6 | 97.76 | 92.55 | 92.33 |
| *WP-M1* | 135.5K | 1.08M | 0.85 | 8 | 98.39 | 95.82 | 95.48 |
| *WP-M2* | 135.5K | 1.08M | 0.85 | 4 | 97.51 | 89.29 | 89.13 |
| *NP-M1* | 135.0K | 1.08M | 0.85 | 8 | 97.86 | 92.42 | 91.90 |
| *NP-M2* | 135.0K | 1.08M | 0.85 | 4 | 97.53 | 92.07 | 91.70 |
| *FS-M1* | 135.0K | 1.08M | 0.85 | 4 | 97.62 | 92.33 | 92.05 |
| *WP-S1* | 100.0K | 0.80M | 0.63 | 5 | 95.32 | 81.64 | 81.66 |
| *WP-S2* | 100.0K | 0.80M | 0.63 | 5 | 95.10 | 72.19 | 71.64 |
| *NP-S1* | 91.3K | 0.73M | 0.57 | 3 | 94.41 | 62.49 | 61.96 |
| *NP-S2* | 91.3K | 0.73M | 0.57 | 3 | 95.21 | 60.54 | 61.86 |
| *FS-S1* | 91.3K | 0.73M | 0.57 | 1 | 96.88 | 90.67 | 90.73 |

(b) FL with Different Pruning Techniques on CIFAR 10 (IID)

| Model | FLOPs | FLOPs Ratio | Space | Space Ratio | $\Gamma_{min}$ | Accuracy |
|-------|-------|-------------|-------|-------------|------|----------|
| FullNets | 653.8K | 1.00 | 512.8K | 1.00 | 10 | 53.63 |
| WP-L1 | 619.6K | 0.94 | 482.3K | 0.94 | 8 | 53.12 |
| FS-L1 | 619.6K | 0.94 | 476.3K | 0.93 | 8 | 53.08 |
| WP-M1 | 587.0K | 0.89 | 451.9K | 0.89 | 6 | 52.66 |
| *WP-M2 | 587.0K | 0.89 | 451.9K | 0.89 | 7 | 52.99 |
| *WP-M3 | 587.0K | 0.89 | 451.9K | 0.89 | 8 | 54.20 |
| FS-M1 | 585.5K | 0.85 | 440.0K | 0.89 | 6 | 51.87 |
| WP-S1 | 553.7K | 0.84 | 421.5K | 0.82 | 4 | 51.69 |
| *WP-S2 | 553.7K | 0.84 | 421.5K | 0.82 | 7 | 52.20 |
| FS-S1 | 551.4K | 0.84 | 403.5K | 0.78 | 4 | 50.96 |

Table 1: Evaluation results on MNIST and CIFAR-10 Dataset

### 4.1 Experiment Settings and Baseline Notations

We focus on three key points in our experiments: (i) the general convergence of FL with heterogeneous models by different pruning strategies, (ii) the impact of minimum coverage index $\Gamma_{\min}$ and (iii) the impact of pruning-induced noise $\delta^2$. The numerical results provide a comprehensive comparison among existing baselines and heterogeneous FL with new pruning strategies in our framework, to validate our theoretical results. Our code implementation can be found at (supplementary files).

To empirically verify the correctness of our theory, we pick FedAvg, and 4 other pruning techniques from the state-of-the-art as baselines – namely FullNets that can be considered as FedAvg [1] without any pruning, "WP" for weights pruning as used in PruneFL[5], "NP" for neuron pruning as used in [47], "FS" for fixed sub-network as used in HeteroFL [3] and "PT" for pruning with a pre-trained mask as used in [4]; for notation and demonstration simplicity. Let $P_m = \frac{\|m\|_0}{|\theta|}$ be the sparsity of mask $m$, e.g., $P_m = 75\%$ for a model when 25 % of its weights are pruned. Due to page limits, we show selected combinations over 4 pruning levels: 60% workers with full model and 40% workers with 75% pruned model; *M.* 40 % workers with full model and 60% workers with 75% pruned model; *S.* 10% workers with full model, 30% workers with 75% pruned model and 60 % workers with 50% pruned models; *SS.* 40% workers with full model and 60 % workers with 50% pruned models.

## 4.2 Numerical Results

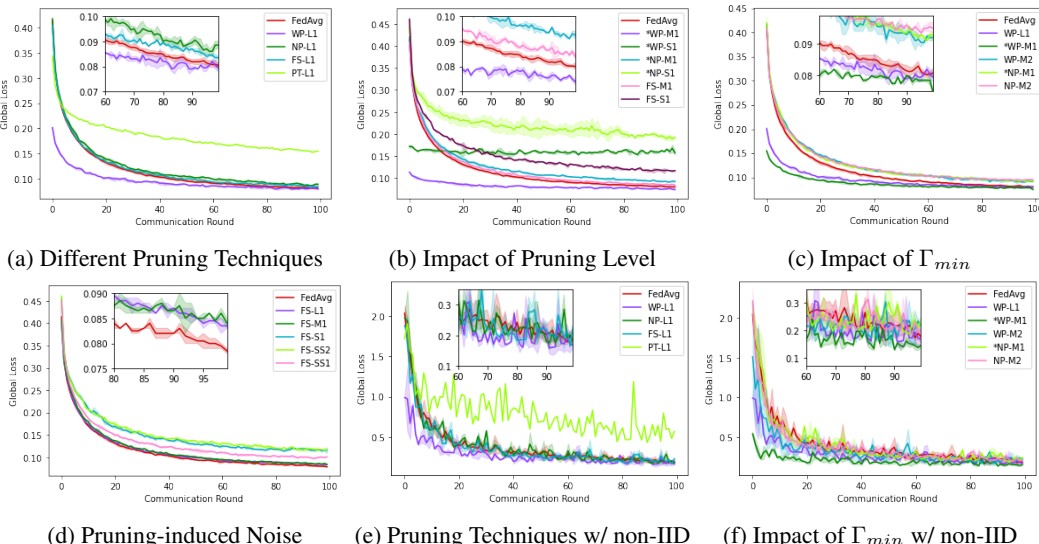

(a) Different Pruning Techniques

(b) Impact of Pruning Level

(c) Impact of $\Gamma_{min}$

(d) Pruning-induced Noise

(e) Pruning Techniques w/ non-IID

(f) Impact of $\Gamma_{min}$ w/ non-IID

Figure 2: Selected experimental results for MNIST (IID and Non-IID) dataset between different pruning settings. (a) For similar pruning level, PT converge a lot slower than others, without a lottery ticket to an optimal mask found. (b) Generally higher pruning level will lead to a higher loss. (c) By applying our design to increase the coverage index, models with identical architecture can reach a solution with lower loss for both selected pruning techniques without additional computational overhead (d) Relative error introduced by pruning is another key factor to the convergence. (e, f)Similar findings are also observed on such scheme with heterogeneous data.

**Impact of minimum coverage index**. Our theory suggests that for a given pruning level (or pruning-induced noise), the minimum coverage index $\Gamma_{min}$ is inverse proportional to the convergence gap as the bound in Theorem 1 indicates. Then for a given pruning level, a pruning strategy in heterogeneous FL with higher minimum coverage index may result in better training performance. Note that existing heterogeneous FL algorithms with adaptive online pruning often focus on removing the small model parameters that are believed to have an insignificant impact on model performance, while being oblivious to the coverage of parameters in pruned local models. Our analysis in this paper illuminates this important design metric for pruning strategies in heterogeneous FL.

To illustrate the importance of minimum coverage index, we consider that the parameters of a model is arg-sorted based on certain pruning technique policy $\mathbb{P}$ and then $K = 4$ regions/partitions are thereby generated representing the highest 25% partition to the lowest 25% partition: $\mathbb{P}_1(\theta) = \{S_1, S_2, S_3, S_4\}$. A pruning mask generated by existing baselines (like in heteroFl and PruneFL) for a 75% sparsity model is then defined as $m_i = 1$ if $\theta_i \in \{S_1 \cup S_2 \cup S_3\}$ otherwise $m_i = 0$. It is easy to see $\Gamma_{min}$ is then directly determined by the number of models with lowest pruning levels, e.g. $\Gamma_{min} = 4$ for Medium pruning level: 40 % workers with full models and 60% workers with 75% pruned models.

We consider a straightforward way to increase the minimum coverage index in the pruning mask design by jointly designing pruning masks for local models to maximize $\Gamma_{min}$. As an example shown in Fig 1, for model with code name *WP-M1, from which 2 out of 6 models with 75% pruned model using regular weights pruning technique, with the other 4 each two use $\mathbb{P}_2(\theta) = \{S_1, S_3, S_4\}$ and $\mathbb{P}_3(\theta) = \{S_1, S_2, S_4\}$, so that $\Gamma_{min} = 8$ is then achieved. We denote such design on current pruning techniques marked with (*) in the results. For detailed case settings, pruning techniques and examples please see Appendix 2 and Appendix 3. As shown in Figure 2(c), under the same model setting with the same pruning level, both pruning techniques with different minimum coverage index show different convergence behaviors. Specifically, the design with a higher minimum coverage index is able to converge faster with lower loss. There are even cases where settings under our design with fewer communication and computation costs that perform better than a design with more costs, e.g. "*WP-M1" over "WP-L1" and "FS-L1" on both IID and non-IID data.

**Impact of pruning-induced noise**. As suggested in our analysis, another key factor that affects convergence is pruning-induced noise $\delta^2$. When a model is pruned, inevitably the pruning-induced

noise $\delta^2$ will affect convergence and model accuracy. Yet, our analysis shows that increasing local epochs or communication rounds cannot mitigate such noise. To minimize the convergence gap in the upperbounds, it is necessary to design pruning strategies in heterogeneous FL with respect to both pruning-induced noise and minimum coverage index, e.g., by considering a joint objective of preserving large parameters while sufficiently covering all parameters. For this phenomenon, we focus on tested higher pruning levels as shown in Figure 2(d) and confirms such a trend. As shown in Figure 2 (b), all selected pruning methods are affected by the change of pruning level.

**More discussions and empirical findings**. In Figure 2(a), PT converges a lot slower than others with its pre-trained masks, as also suggested by previous works that models with static sparse parameters will converge to a solution with higher loss than models with dynamic sparse training. Finally, we also show a synthetic special case where all local clients do not sum up a mask that covers the whole model (proposed conditions are not met) in Appendix.4, where the it did not learn a usable solution.

## 5 Conclusion

In this paper, we study the sufficient conditions for FL with heterogeneous local models – which may vary over time and across clients – to converge to a small neighborhood of a stationary point of standard FL, at a rate of $\frac{1}{\sqrt{Q}}$. The optimality gap is characterized and depends on pruning-induced noise and a new notion of minimum coverage index. The result recovers a number of important FL algorithms as special cases. It also provides new insights on designing optimized pruning strategies in heterogeneous FL, with respect to both minimum coverage index $\Gamma_{\min}$ and pruning-induced noise $\delta^2$. We empirically demonstrated the correctness of the theory and the design insights. Our work provides a theoretical understanding of heterogeneous FL with adaptive local model pruning and presents valuable insights on new algorithm design, which will be considered in future work.

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

# A Proof of Theorems

## A.1 Problem summary and notations

We summarize the algorithm in a way that can present the convergence analysis more easily. We use a superscript such as $\theta^{(i)}$, $m_{q,n}^{(i)}$, and $\nabla F^{(i)}$ to denote the sub-vector of parameter, mask, and gradient corresponding to region $i$. For the proof purpose and with slight abuse of notations, we denote all modeling parameters contained in the same set of local models as a parameter region $i$ (Ultimately we can regard each modeling parameter as a separate region). In each round $q$, parameters in each region $i$ is contained in and only in a set of local models denoted by $\mathcal{N}_q^{(i)}$, implying that $m_{q,n}^{(i)} = \mathbf{1}$ for $n \in \mathcal{N}_q^{(i)}$ and $m_{q,n}^{(i)} = \mathbf{0}$ otherwise, for all the parameters in the region. We define $\Gamma^* = \min_{q,i} \mathcal{N}_q^{(i)}$ as the minimum coverage index, since it denotes the minimum number of local models that contain any parameters in $\theta_q$. With slight abuse of notations, we use $\nabla F_n(\theta$ and $\nabla F_n(\theta, \xi)$ to denote the the gradient and stochastic gradient, respectively.

---

**Algorithm 1:** Heterogeneous FL with adaptive online model pruning

---

**Input:** Local data $D_i^k$ on $N$ local workers, learning rate $\gamma$, pruning policy $\mathbb{P}$, number of local epochs $T$, global model parameterized by $\theta$.

**Executes:**

Initialize $\theta_0$

**for** *round $q = 1, 2, \ldots, Q$* **do do**

    **for** *local workers $n = 1, 2, \ldots, N$* **do** *(In parallel)* **do**

        Generate mask $m_{q,n} = \mathbb{P}(\theta_q, n)$

        Prune $\theta_{q,n,0} = \theta_q \odot m_{q,n}$

        // Update local models:

        **for** *epoch $t = 1, 2, \ldots, T$* **do do**

            Update $\theta_{q,n,t} = \theta_{q,n,t-1} - \gamma \nabla F_n(\theta_{q,n,t-1}, \xi_{n,t-1}) \odot m_{q,n}$

        **end**

    **end**

    // Update global model:

    **for** *region $i = 1, 2, \ldots, K$* **do do**

        Find $\mathcal{N}_q^{(i)} = \{n : m_{q,n}^{(i)} = \mathbf{1}\}$

        Update $\theta_{q+1}^{(i)} = \frac{1}{|\mathcal{N}_q^{(i)}|} \sum_{n \in \mathcal{N}_q^{(i)}} \theta_{q,n,T}^{(i)}$

    **end**

**end**

Output $\theta_Q$

---

## A.2 Assumptions

**Assumption 1.** *(Smoothness). Cost functions $F_1, \ldots, F_N$ are all L-smooth: $\forall \theta, \phi \in \mathcal{R}^d$ and any $n$, we assume that there exists $L > 0$:*

$$\|\nabla F_n(\theta) - \nabla F_n(\phi)\| \le L\|\theta - \phi\|. \tag{1}$$

**Assumption 2.** *(Pruning-induced Error). We assume that for some $\delta^2 \in [0, 1)$ and any $q, n, t$, the pruning-induced error is bounded by*

$$\|\theta_{q,n,t} - \theta_{q,n,t} \odot m_{q,n}\|^2 \le \delta^2 \|\theta_{q,n,t}\|^2. \tag{2}$$

**Assumption 3.** *(Bounded Gradient). The expected squared norm of stochastic gradients is bounded uniformly, i.e., for constant $G > 0$ and any $n, q, t$:*

$$E \|\nabla F_n(\theta_{q,n,t}, x_{q,n,t})\|^2 \le G. \tag{3}$$

**Assumption 4.** *(Gradient Noise for IID data). Under IID data distribution, for any $q, n, t$, we assume that*

$$\mathbb{E}[\nabla F_n(\theta_{q,n,t}, \xi_{n,t})] = \nabla F(\theta_{q,n,t}) \tag{4}$$

$$\mathbb{E}\|\nabla F_n(\theta_{q,n,t}, \xi_{n,t}) - \nabla F(\theta_{q,n,t})\|^2 \le \sigma^2 \tag{5}$$

*where $\sigma^2 > 0$ is a constant and $\xi_{n,t})$ are independent samples for different $n, t$.*

**Assumption 5.** *(Gradient Noise for non-IID data). Under non-IID data distribution, we assume that for constant $\sigma^2 > 0$ and any $q, n, t$:*

$$\mathbb{E}\left[\frac{1}{|\mathcal{N}_q^{(i)}|}\sum_{n\in\mathcal{N}_q^{(i)}}\nabla F_n^{(i)}(\theta_{q,n,t},\xi_{n,t})\right] = \nabla F^{(i)}(\theta_{q,n,t}) \tag{6}$$

$$\mathbb{E}\left\|\frac{1}{|\mathcal{N}_q^{(i)}|}\sum_{n\in\mathcal{N}_q^{(i)}}\nabla F_n^{(i)}(\theta_{q,n,t},\xi_{n,t}) - \nabla F^{(i)}(\theta_{q,n,t})\right\|^2 \le \sigma^2. \tag{7}$$

### A.3 Convergence Analysis

We now analyze the convergence of heterogeneous FL under adaptive online model pruning with respect to any pruning policy $\mathbb{P}(\theta_q, n)$ (and the resulting mask $m_{q,n}$) and prove the main theorems in this paper. We need to overcome a number of challenges as follows:

- We will begin the proof by analyzing the change of loss function in one round as the model goes from $\theta_q$ to $\theta_{q+1}$, i.e., $F(\theta_{q+1}) - F(\theta_1)$. It includes three major steps: pruning to obtain heterogeneous local models $\theta_{q,n,0} = \theta_q \odot m_{q,n}$, training local models in a distributed fashion to update $\theta_{q,n,t}$, and parameter aggregation to update the global model $\theta_{q+1}$.
- Due to the use of heterogeneous local models whose masks $m_{q,n}$ both vary over rounds and change for different workers, we first characterize the difference between local model $\theta_{q,n,t}$ at any epoch $t$ and global model $\theta_q$ at the beginning of the current round. It is easy to see that this can be factorized into two parts: pruning induced error $\|\theta_{q,n,0} - \theta_q\|^2$ and local training $\|\theta_{q,n,t} - \theta_{q,n,0}\|^2$, which will be analyzed in Lemma 1.
- We characterize the impact of heterogeneous local models on global parameter update. Specifically, we use an ideal local gradient $\nabla F_n(\theta_q)$ as a reference point and quantify the different between aggregated local gradients and the ideal gradient. This will be presented in Lemma 2. We also quantify the norm difference between a gradient and a stochastic gradient (with respect to the global update step) using the gradient noise assumptions, in Lemma 3.
- Since IID and non-IID data distributions in our model differ in the gradient noise assumption (i.e., Assumption 4 and Assumption 5), we present a unified proof for both cases. We will explicitly state IID and non-IID data distributions only if the two cases require different treatment (when the gradient noise assumptions are needed). Otherwise, the derivations and proofs are identical for both cases.

We will begin by proving a number of lemmas and then use them for convergence analysis.

**Lemma 1.** *Under Assumption 2 and Assumption 3, for any $q$, we have:*

$$\sum_{t=1}^{T}\sum_{n=1}^{N}\mathbb{E}\|\theta_{q,n,t-1} - \theta_q\|^2 \le \gamma^2 T^2 NG + \delta^2 NT \cdot \mathbb{E}\|\theta_q\|^2. \tag{8}$$

*Proof.* We note that $\theta_q$ is the global model at the beginning of current round. We split the difference $\theta_{q,n,t-1} - \theta_q$ into two parts: changes due to local model training $\theta_{q,n,t-1} - \theta_{q,n,0}$ and changes due to pruning $\theta_{q,n,0} - \theta_q$. That is

$$\sum_{t=1}^{T}\sum_{n=1}^{N}\mathbb{E}\|\theta_{q,n,t-1} - \theta_q\|^2$$

$$= \sum_{t=1}^{T}\sum_{n=1}^{N}\mathbb{E}\|\left(\theta_{q,n,t-1} - \theta_{q,n,0}\right) + \left(\theta_{q,n,0} - \theta_q\right)\|^2$$

$$\le \sum_{t=1}^{T}\sum_{n=1}^{N}2\mathbb{E}\|\theta_{q,n,t-1} - \theta_q\|^2 + \sum_{t=1}^{T}\sum_{n=1}^{N}2\mathbb{E}\|\theta_{q,n,t-1} - \theta_q\|^2 \tag{9}$$

where we used the fact that $\|\sum_{i=1}^{s} a_i\|^2 \le s\sum_{i=1}^{s}\|a_i\|^2$ in the last step.

For the first term in Eq.(9), we notice that $\theta_{q,n,t-1}$ is obtained from $\theta_{q,n,0}$ through $t-1$ epochs of local model updates on worker $n$. Using the local gradient updates from the algorithm, it is easy to

see:

$$\sum_{t=1}^{T} \sum_{n=1}^{N} \mathbb{E}\|\theta_{q,n,t-1} - \theta_{q,n,0}\|^2$$

$$= \sum_{t=1}^{T} \sum_{n=1}^{N} \mathbb{E} \left\| \sum_{j=0}^{t-1} -\gamma \nabla F_n(\theta_{q,n,t-1}, \xi_{n,t-1}) \odot m_{q,n} \right\|^2$$

$$\leq \sum_{t=1}^{T} \sum_{n=1}^{N} (t-1) \sum_{j=0}^{t-1} \mathbb{E} \left\| -\gamma \nabla F_n(\theta_{q,n,t-1}, \xi_{n,t-1}) \odot m_{q,n} \right\|^2$$

$$\leq \sum_{t=1}^{T} \sum_{n=1}^{N} (t-1)\gamma^2 G$$

$$\leq \frac{\gamma^2 T^2 N G}{2}, \tag{10}$$

where we use the fact that $\|\sum_{i=1}^{s} a_i\|^2 \leq s \sum_{i=1}^{s} \|a_i\|^2$ in step 2 above, and the fact that $m_{q,n}$ is a binary mask in step 3 above together with Assumption 3 for bounded gradient.

For the second term in Eq.(9), the difference is resulted by model pruning using mask $m_{n,q}$ of work $n$ in round $q$. We have

$$\sum_{t=1}^{T} \sum_{n=1}^{N} \mathbb{E}\|\theta_{q,n,0} - \theta_q\|^2 \quad = \sum_{t=1}^{T} \sum_{n=1}^{N} \mathbb{E}\|\theta_q \odot m_{n,q} - \theta_q\|^2$$

$$\leq \sum_{t=1}^{T} \sum_{n=1}^{N} \delta^2 \mathbb{E}\|\theta_q\|^2$$

$$= \delta^2 NT \cdot \mathbb{E}\|\theta_q\|^2, \tag{11}$$

where we used the fact that $\theta_{q,n,0} = \theta_q \odot m_{n,q}$ in step 1 above, and Assumption 2 in step 2 above.

Plugging Eq.(10) and Eq.(11) into Eq.(9), we obtain the desired result. $\qquad\square$

**Lemma 2.** *Under Assumptions 1-3, for any $q$, we have:*

$$\sum_{i=1}^{K} \mathbb{E} \left\| \frac{1}{\Gamma_q^{(i)} T} \sum_{t=1}^{T} \sum_{n \in \mathcal{N}_q^{(i)}} \left[ \nabla F_n^{(i)}(\theta_{q,n,t-1}) - \nabla F_n^{(i)}(\theta_q) \right] \right\|^2$$

$$\leq \frac{L^2 \gamma^2 TNG}{\Gamma^*} + \frac{L^2 \delta^2 N}{\Gamma^*} \mathbb{E}\|\theta_q\|^2. \tag{12}$$

*Proof.* Recall that $\Gamma_q^{(i)} = |\mathcal{N}_q^{(i)}|$ is the number of local models containing parameters of region $i$ in round $q$. The left-hand-side of Eq.(12) denotes the difference between an average gradient of heterogeneous models (through aggregation and over time) and an ideal gradient. The summation over $i$ adds up such difference over all regions $i = 1, \ldots, K$, because the average gradient takes a different form in different regions.

From the inequality $\| \sum_{i=1}^{s} a_i \|^2 \le s \sum_{i=1}^{s} \|a_i\|^2$, we obtain $\| \frac{1}{s} \sum_{i=1}^{s} a_i \|^2 \le \frac{1}{s} \sum_{i=1}^{s} \|a_i\|^2$. We use this inequality on the left-hand-side of Eq.(12) to get:

$$
\begin{aligned}
\sum_{i=1}^{K} \mathbb{E} & \left\| \frac{1}{\Gamma_q^{(i)} T} \sum_{t=1}^{T} \sum_{n \in \mathcal{N}_q^{(i)}} \left[ \nabla F_n^{(i)}(\theta_{q,n,t-1}) - \nabla F_n^{(i)}(\theta_q) \right] \right\|^2 \\
& \le \sum_{i=1}^{K} \frac{1}{\Gamma_q^{(i)} T} \sum_{t=1}^{T} \sum_{n \in \mathcal{N}_q^{(i)}} \mathbb{E} \left\| \nabla F_n^{(i)}(\theta_{q,n,t-1}) - \nabla F_n^{(i)}(\theta_q) \right\|^2 \\
& \le \frac{1}{T\Gamma^*} \sum_{t=1}^{T} \sum_{n=1}^{N} \sum_{i=1}^{K} \mathbb{E} \left\| \nabla F_n^{(i)}(\theta_{q,n,t-1}) - \nabla F_n^{(i)}(\theta_q) \right\|^2 \\
& = \frac{1}{T\Gamma^*} \sum_{t=1}^{T} \sum_{n=1}^{N} \mathbb{E} \left\| \nabla F_n(\theta_{q,n,t-1}) - \nabla F_n(\theta_q) \right\|^2 \\
& \le \frac{1}{T\Gamma^*} \sum_{t=1}^{T} \sum_{n=1}^{N} L^2 \mathbb{E} \left\| \theta_{q,n,t-1} - \theta_q \right\|^2 ,
\end{aligned}
\tag{13}
$$

where we relax the inequality by choosing the smallest $\Gamma^* = \min_{q,i} \Gamma_q^{(i)}$ and changing the summation over $n$ to all workers in the second step. In the third step, we use the fact that $L_2$ gradient norm of a vector is equal to the sum of norm of all sub-vectors (i.e., regions $i = 1, \dots, K$). This allows us to consider $\nabla F_n$ instead of its sub-vectors on different regions.

Finally, the last step is directly from L-smoothness in Assumption 1. Under Assumptions 2-3, we notice that the last step of Eq.(13) is further bounded by Lemma 1, which yields the desired result of this lemma after re-arranging the terms. □

**Lemma 3.** *For IID data distribution under Assumptions 4, for any $q$, we have:*

$$
\sum_{i=1}^{K} \mathbb{E} \left\| \frac{1}{\Gamma_q^{(i)} T} \sum_{t=1}^{T} \sum_{n \in \mathcal{N}_q^{(i)}} \left[ \nabla F_n^{(i)}(\theta_{q,n,t-1}, \xi_{n,t-1}) - \nabla F^{(i)}(\theta_{q,n,t-1}) \right] \right\|^2 \le \frac{N\sigma^2}{T(\Gamma^*)^2}.
$$

*For non-IID data distribution under Assumption 5, for any $q$, we have:*

$$
\sum_{i=1}^{K} \mathbb{E} \left\| \frac{1}{\Gamma_q^{(i)} T} \sum_{t=1}^{T} \sum_{n \in \mathcal{N}_q^{(i)}} \left[ \nabla F_n^{(i)}(\theta_{q,n,t-1}, \xi_{n,t-1}) - \nabla F^{(i)}(\theta_{q,n,t-1}) \right] \right\|^2 \le \frac{K\sigma^2}{T}.
$$

*Proof.* This lemma quantifies the square norm of the difference between gradient and stochastic gradient in the global parameter update. We present results for both IID and non-IID cases in this lemma under Assumption 4 and Assumption 5, respectively.

We first consider IID data distributions. Since all the samples $\xi_{n,t-1}$ are independent from each other for different $n$ and $t-1$, the difference between gradient and stochastic gradient, i.e., $\nabla F_n^{(i)}(\theta_{q,n,t-1}, \xi_{n,t-1}) - \nabla F_n^{(i)}(\theta_{q,n,t-1})$, are independent gradient noise. Due to Assumption 4, these gradient noise has zero mean. Using the fact that $\mathbb{E}\| \sum_i \mathbf{x}_i \|^2 = \sum_i \mathbb{E}\|\mathbf{x}_i^2\|$ for zero-mean and

independent $\mathbf{x}_i$'s, we get:

$$\sum_{i=1}^{K} \mathbb{E} \left\| \frac{1}{\Gamma_q^{(i)} T} \sum_{t=1}^{T} \sum_{n \in \mathcal{N}_q^{(i)}} \left[ \nabla F_n^{(i)}(\theta_{q,n,t-1}, \xi_{n,t-1}) - \nabla F_n^{(i)}(\theta_{q,n,t-1}) \right] \right\|^2$$

$$\leq \sum_{i=1}^{K} \frac{1}{(\Gamma_q^{(i)} T)^2} \sum_{t=1}^{T} \sum_{n \in \mathcal{N}_q^{(i)}} \mathbb{E} \left\| \nabla F_n^{(i)}(\theta_{q,n,t-1}, \xi_{n,t-1}) - \nabla F_n^{(i)}(\theta_{q,n,t-1}) \right\|^2$$

$$\leq \frac{1}{(T\Gamma^*)^2} \sum_{i=1}^{K} \sum_{t=1}^{T} \sum_{n=1}^{N} \mathbb{E} \left\| \nabla F_n^{(i)}(\theta_{q,n,t-1}, \xi_{n,t-1}) - \nabla F_n^{(i)}(\theta_{q,n,t-1}) \right\|^2$$

$$= \frac{1}{(T\Gamma^*)^2} \sum_{t=1}^{T} \sum_{n=1}^{N} \mathbb{E} \left\| \nabla F_n(\theta_{q,n,t-1}, \xi_{n,t-1}) - \nabla F_n(\theta_{q,n,t-1}) \right\|^2$$

$$\leq \frac{1}{(T\Gamma^*)^2} \cdot TN\sigma^2 \tag{14}$$

where we used the property of zero-mean and independent gradient noise in the first step above, relax the inequality by choosing the smallest $\Gamma^* = \min_{q,i} \Gamma_q^{(i)}$ and changing the summation over $n$ to all workers in the second step. In the third step, we use the fact that $L_2$ gradient norm of a vector is equal to the sum of norm of all sub-vectors (i.e., regions $i = 1, \ldots, K$). This allows us to consider $\nabla F_n$ instead of its sub-vectors on different regions. Finally, we apply Assumption 4 to bound the gradient noise and obtain the desired result.

For non-IID data distributions under Assumption 4 (instead of Assumption 5), we notice that $\mathbb{E}\left[ \frac{1}{|\mathcal{N}_q^{(i)}|} \sum_{n \in \mathcal{N}_q^{(i)}} \nabla F_n^{(i)}(\theta_{q,n,t-1}, \xi_{n,t-1}) \right] = \nabla F^{(i)}(\theta_{q,n,t-1})$ is an unbiased estimate for any epoch $t$, with bounded gradient noise. Again, due to independent samples $\xi_{n,t-1}$, we have:

$$\sum_{i=1}^{K} \mathbb{E} \left\| \frac{1}{\Gamma_q^{(i)} T} \sum_{t=1}^{T} \sum_{n \in \mathcal{N}_q^{(i)}} \left[ \nabla F_n^{(i)}(\theta_{q,n,t-1}, \xi_{n,t-1}) - \nabla F_n^{(i)}(\theta_{q,n,t-1}) \right] \right\|^2$$

$$\leq \frac{1}{T^2} \sum_{i=1}^{K} \sum_{t=1}^{T} \mathbb{E} \left\| \frac{1}{\Gamma_q^{(i)}} \sum_{n \in \mathcal{N}_q^{(i)}} \nabla F_n^{(i)}(\theta_{q,n,t-1}, \xi_{n,t-1}) - \nabla F_n^{(i)}(\theta_{q,n,t-1}) \right\|^2$$

$$\leq \frac{1}{T^2} \sum_{i=1}^{K} \sum_{t=1}^{T} \sigma^2$$

$$= \frac{K\sigma^2}{T}, \tag{15}$$

where we use the property of zero-mean and independent gradient noise in the first step above, used the fact that the norm of a sub-vector (in region $i$) is bounded by that of the entire vector in the second step above, as well as Assumption 5. This completes the proof of this lemma. $\square$

**Proof of the main result**. Now we are ready to present the main proof. We begin with the L-smoothness property in Assumption 1, which implies

$$F(\theta_{q+1}) - F(\theta_q) \leq \langle \nabla F(\theta_q), \; \theta_{q+1} - \theta_q \rangle + \frac{L}{2} \|\theta_{q+1} - \theta_q\|^2. \tag{16}$$

We take expectations on both sides of the inequality and get:

$$\mathbb{E}[F(\theta_{q+1})] - \mathbb{E}[]F(\theta_q)] \leq \mathbb{E} \langle \nabla F(\theta_q), \; \theta_{q+1} - \theta_q \rangle + \frac{L}{2} \mathbb{E} \|\theta_{q+1} - \theta_q\|^2. \tag{17}$$

In the following, we bound the two terms on the right-hand-side above and finally combine the results to complete the proof.

**Upperbound for** $\mathbb{E}\langle \nabla F(\theta_q),\, \theta_{q+1} - \theta_q \rangle$. We notice that the inner product can be broken down and reformulated as the sum of inner products over all regions $i = 1, \ldots, K$. This is necessary because the global parameter update is different for different regions. More precisely, for any region $i$, we have:

$$
\begin{aligned}
\theta_{q+1}^{(i)} - \theta_q^{(i)} &= \left( \frac{1}{\Gamma_q^{(i)}} \sum_{n \in \mathcal{N}_q^{(i)}} \theta_{q,n,T}^{(i)} \right) - \theta_q^{(i)} \\
&= \frac{1}{\Gamma_q^{(i)}} \sum_{n \in \mathcal{N}_q^{(i)}} \left[ \theta_{q,n,0}^{(i)} - \sum_{t=1}^{T} \gamma \nabla F_n^{(i)}(\theta_{q,n,t-1}, \xi_{n,t-1}) \cdot m_{n,q}^{(i)} \right] - \theta_q^{(i)} \\
&= -\frac{1}{\Gamma_q^{(i)}} \sum_{n \in \mathcal{N}_q^{(i)}} \sum_{t=1}^{T} \gamma \nabla F_n^{(i)}(\theta_{q,n,t-1}, \xi_{n,t-1}) \cdot m_{n,q}^{(i)} + \theta_q^{(i)} \cdot m_{n,q}^{(i)} - \theta_q^{(i)} \\
&= -\frac{1}{\Gamma_q^{(i)}} \sum_{n \in \mathcal{N}_q^{(i)}} \sum_{t=1}^{T} \gamma \nabla F_n^{(i)}(\theta_{q,n,t-1}, \xi_{n,t-1}),
\end{aligned}
\tag{18}
$$

where global parameter updated is used in the first step, local parameter update is used in the second step, and the third step follows from the fact that for any worker $n \in \mathcal{N}_q^{(i)}$ participating in the global update of $\theta_q^{(i)}$ contain the model parameters of region $i$, i.e., $m_{q,n}^{(i)} = \mathbf{1}$. We also use $\theta_{q,n,0}^{(i)} = \theta_q^{(i)} \cdot m_{n,q}^{(i)}$ in the third step above because of to pruning.

Next we analyze $\mathbb{E}\langle \nabla F(\theta_q),\, \theta_{q+1} - \theta_q \rangle$ by considering a sum of inner products over $K$ regions. We have

$$
\begin{aligned}
&\mathbb{E}\langle \nabla F(\theta_q),\, \theta_{q+1} - \theta_q \rangle \\
&= \sum_{i=1}^{K} \mathbb{E}\left\langle \nabla F^{(i)}(\theta_q),\, \theta_{q+1}^{(i)} - \theta_q^{(i)} \right\rangle \\
&= \sum_{i=1}^{K} \mathbb{E}\left\langle \nabla F^{(i)}(\theta_q),\, -\frac{1}{\Gamma_q^{(i)}} \sum_{n \in \mathcal{N}_q^{(i)}} \sum_{t=1}^{T} \gamma \nabla F_n^{(i)}(\theta_{q,n,t-1}, \xi_{n,t-1}) \right\rangle \\
&= \sum_{i=1}^{K} \mathbb{E}\left\langle \nabla F^{(i)}(\theta_q),\, -\frac{1}{\Gamma_q^{(i)}} \sum_{n \in \mathcal{N}_q^{(i)}} \sum_{t=1}^{T} \gamma \mathbb{E}\left[ \nabla F_n^{(i)}(\theta_{q,n,t-1}, \xi_{n,t-1}) | \theta_q \right] \right\rangle \\
&= \sum_{i=1}^{K} \mathbb{E}\left\langle \nabla F^{(i)}(\theta_q),\, -\frac{1}{\Gamma_q^{(i)}} \sum_{n \in \mathcal{N}_q^{(i)}} \sum_{t=1}^{T} \gamma \nabla F_n^{(i)}(\theta_{q,n,t-1}) \right\rangle \\
&= -\sum_{i=1}^{K} \mathbb{E}\left\langle \nabla F^{(i)}(\theta_q),\, \gamma T \nabla F^{(i)}(\theta_q) \right\rangle \\
&\quad - \sum_{i=1}^{K} \mathbb{E}\left\langle \nabla F^{(i)}(\theta_q),\, \frac{1}{\Gamma_q^{(i)}} \sum_{n \in \mathcal{N}_q^{(i)}} \sum_{t=1}^{T} \gamma \left[ \nabla F_n^{(i)}(\theta_{q,n,t-1}) - \nabla F^{(i)}(\theta_q) \right] \right\rangle
\end{aligned}
\tag{19}
$$

where we use the first step reformulates the inner product as a sum, the second step follows from Eq.(18), the third step employs a conditional expectation over the random samples with respect to $\theta_q$, and the last step splits the result into two parts with respect to a reference point $\gamma T \nabla F^{(i)}(\theta_q)$.

For the first term on the right-hand-side of Eq.(19), it is easy to see that

$$
\begin{aligned}
-\sum_{i=1}^{K} \mathbb{E}\left\langle \nabla F^{(i)}(\theta_q),\, \gamma T \nabla F^{(i)}(\theta_q) \right\rangle &= -\gamma T \sum_{i=1}^{K} \left\| \nabla F^{(i)}(\theta_q) \right\|^2 \\
&= -\gamma T \left\| \nabla F(\theta_q) \right\|^2,
\end{aligned}
\tag{20}
$$

where we add up the norm over $K$ regions in the last step. For the second term on the right-hand-side of Eq.(19), we use the inequality $< a, b > \le \frac{1}{2}\|a\|^2 + \frac{1}{2}\|b\|^2$ for any vectors $a, b$. Applying this inequality to the second term, we have

$$-\sum_{i=1}^{K} \mathbb{E} \left\langle \nabla F^{(i)}(\theta_q), \frac{1}{\Gamma_q^{(i)}} \sum_{n \in \mathcal{N}_q^{(i)}} \sum_{t=1}^{T} \gamma \left[ \nabla F_n^{(i)}(\theta_{q,n,t-1}) - \nabla F^{(i)}(\theta_q) \right] \right\rangle$$

$$= -\sum_{i=1}^{K} T\gamma \cdot \mathbb{E} \left\langle \nabla F^{(i)}(\theta_q), \frac{1}{T\Gamma_q^{(i)}} \sum_{n \in \mathcal{N}_q^{(i)}} \sum_{t=1}^{T} \left[ \nabla F_n^{(i)}(\theta_{q,n,t-1}) - \nabla F^{(i)}(\theta_q) \right] \right\rangle$$

$$\le \frac{T\gamma}{2} \sum_{i=1}^{K} \mathbb{E} \left\| \nabla F^{(i)}(\theta_q) \right\|^2 + \frac{T\gamma}{2} \sum_{i=1}^{K} \mathbb{E} \left\| \frac{1}{T\Gamma_q^{(i)}} \sum_{n \in \mathcal{N}_q^{(i)}} \sum_{t=1}^{T} \left[ \nabla F_n^{(i)}(\theta_{q,n,t-1}) - \nabla F^{(i)}(\theta_q) \right] \right\|$$

$$= \frac{T\gamma}{2} \mathbb{E} \|\nabla F(\theta_q)\|^2 + \frac{T\gamma}{2} \left( \frac{L^2 \gamma^2 T N G}{\Gamma^*} + \frac{L^2 \delta^2 N}{\Gamma^*} \mathbb{E}\|\theta_q\|^2 \right) \tag{21}$$

where the second step uses the inequality and the third step follows directly from Lemma 2. Plugging Eq.(20) and Eq.(21) results into Eq.(19), we obtain the desired upperbound:

$$\mathbb{E} \langle \nabla F(\theta_q), \theta_{q+1} - \theta_q \rangle \le -\frac{T\gamma}{2} \mathbb{E} \|\nabla F(\theta_q)\|^2 + \frac{T\gamma}{2} \left( \frac{L^2 \gamma^2 T N G}{\Gamma^*} + \frac{L^2 \delta^2 N}{\Gamma^*} \mathbb{E}\|\theta_q\|^2 \right). \tag{22}$$

**Upperbound for** $\frac{L}{2}\mathbb{E}\|\theta_{q+1} - \theta_q\|^2$. We use the again result in Eq.(18) and apply it to $\theta_{q+1} - \theta_q$, which gives:

$$\frac{L}{2}\mathbb{E}\|\theta_{q+1} - \theta_q\|^2$$

$$= \frac{L}{2}\mathbb{E} \left\| \frac{1}{\Gamma_q^{(i)}} \sum_{n \in \mathcal{N}_q^{(i)}} \sum_{t=1}^{T} \gamma \nabla F_n^{(i)}(\theta_{q,n,t-1}, \xi_{n,t-1}) \right\|^2$$

$$\le \frac{3L}{2}\mathbb{E} \left\| \frac{1}{\Gamma_q^{(i)}} \sum_{n \in \mathcal{N}_q^{(i)}} \sum_{t=1}^{T} \gamma \left[ \nabla F_n^{(i)}(\theta_{q,n,t-1}, \xi_{n,t-1}) - \nabla F_n^{(i)}(\theta_{q,n,t-1}) \right] \right\|^2$$

$$+ \frac{3L}{2}\mathbb{E} \left\| \frac{1}{\Gamma_q^{(i)}} \sum_{n \in \mathcal{N}_q^{(i)}} \sum_{t=1}^{T} \gamma \left[ \nabla F_n^{(i)}(\theta_{q,n,t-1}) - \nabla F_n^{(i)}(\theta_q) \right] \right\|^2$$

$$+ \frac{3L}{2}\mathbb{E} \left\| \frac{1}{\Gamma_q^{(i)}} \sum_{n \in \mathcal{N}_q^{(i)}} \sum_{t=1}^{T} \gamma \nabla F_n^{(i)}(\theta_q) \right\|^2, \tag{23}$$

where in the second step, we use the inequality $\|\sum_{i=1}^{s} a_i\|^2 \le s \sum_{i=1}^{s} \|a_i\|^2$ and split stochastic gradient $[\nabla F_n^{(i)}(\theta_{q,n,t-1}, \xi_{n,t-1})]$ into $s = 3$ parts, i.e., $[\nabla F_n^{(i)}(\theta_{q,n,t-1}, \xi_{n,t-1}) - \nabla F_n^{(i)}(\theta_{q,n,t-1})]$, $[F_n^{(i)}(\theta_{q,n,t-1}) - F_n^{(i)}(\theta_q)]$, and $[F_n^{(i)}(\theta_q)]$.

Next, we notice that the third term on the right-hand-side of Eq.(23) can be simplified, because (i) for IID data distribution, the cost function of each worker $n$ is the same as the global cost function, i.e., $\nabla F_n(\theta_q) = \nabla F(\theta_q)$, and (ii) for non-IID data distribution, the gradient noise assumption (Assumption 5) implies that $\frac{1}{\Gamma_q^{(i)}} \sum_{n \in \mathcal{N}_q^{(i)}} \nabla F_n(\theta_q) = F(\theta_q)$. Thus in both cases, we have:

$$\frac{3L}{2}\mathbb{E} \left\| \frac{1}{\Gamma_q^{(i)}} \sum_{n \in \mathcal{N}_q^{(i)}} \sum_{t=1}^{T} \gamma \nabla F_n^{(i)}(\theta_q) \right\|^2 \le \frac{3LT^2\gamma^2}{2} \sum_{i=1}^{K} \mathbb{E}\|\nabla F^{(i)}(\theta_q)\|^2$$

$$= \frac{3LT^2\gamma^2}{2} \mathbb{E}\|F(\theta_q)\|^2, \tag{24}$$

where we again used the sum of norm of $K$ regions in the last step.

Now we notice that the first and second terms of Eq.(23) have been bounded by Lemma 2 and Lemma 3, excpet for constants $\gamma$ and $1/T$. Applying these results directly and also plugging in Eq.(24) into Eq.(23), we obtain the desired upperbound:

$$
\begin{aligned}
\frac{L}{2}\mathbb{E}\left\|\theta_{q+1}-\theta_q\right\|^2 \quad \leq & \frac{3LTN\gamma^2\sigma^2}{2(\Gamma^*)^2}\ (\text{for IID}) \text{ or } \frac{3LTK\gamma^2\sigma^2}{2}\ (\text{for non}-\text{IID}) \\
& +\frac{3L^3\gamma^4T^3NG}{2\Gamma^*}+\frac{3L^3T^2\gamma^2\delta^2N}{2\Gamma^*}\mathbb{E}\|\theta_q\|^2 \\
& +\frac{3LT^2\gamma^2}{2}\mathbb{E}\|F_n(\theta_q)\|^2.
\end{aligned}
\tag{25}
$$

**Combining the two Upperbounds.** Finally, we will apply the upperbound for $\mathbb{E}\left\langle\nabla F(\theta_q),\ \theta_{q+1}-\theta_q\right\rangle$ in Eq.(22) as well as the upperbound for $\frac{L}{2}\mathbb{E}\left\|\theta_{q+1}-\theta_q\right\|^2$ in Eq.(25), and plug them into Eq.(17). First we take the sum over $q=1,\ldots,Q$ on both sides of Eq.(17), which becomes:

$$
\begin{aligned}
&\mathbb{E}[F(\theta_{Q+1})]-\mathbb{E}[F(\theta_0)] \\
&=\sum_{q=1}^{Q}\mathbb{E}[F(\theta_{q+1})]-\sum_{q=1}^{Q}\mathbb{E}[F(\theta_q)] \\
&\leq\sum_{q=1}^{Q}\mathbb{E}\left\langle\nabla F(\theta_q),\ \theta_{q+1}-\theta_q\right\rangle+\sum_{q=1}^{Q}\frac{L}{2}\mathbb{E}\left\|\theta_{q+1}-\theta_q\right\|^2.
\end{aligned}
\tag{26}
$$

Now plugging in the two upperbounds and re-arranging the terms, for IID data distribution, we derive:

$$
\begin{aligned}
&\mathbb{E}[F(\theta_{Q+1})]-\mathbb{E}[F(\theta_0)] \\
&\leq -\frac{T\gamma}{2}\left(1-3LT\gamma\right)\sum_{q=1}^{Q}\mathbb{E}\|\nabla F(\theta_q)\|^2 \\
&\quad +\frac{\gamma TQ}{2}\left(\frac{TL^2\gamma^2NG}{\Gamma^*}+\frac{3LN\gamma\sigma^2}{(\Gamma^*)^2}+\frac{3L^3\gamma^3T^3NG}{\Gamma^*}\right) \\
&\quad +\frac{T\gamma}{2}\left(\frac{L^2\delta^2N}{\Gamma^*}+\frac{3L^3T\gamma\delta^2N}{\Gamma^*}\right)\sum_{q=1}^{T}\mathbb{E}\|\theta_q\|^2.
\end{aligned}
\tag{27}
$$

We choose learning rate $\gamma\leq 1/(6LT)$ and use the fact that $\mathbb{E}[F(\theta_{Q+1})]$ is non-negative. The inequality above becomes:

$$
\begin{aligned}
\frac{T\gamma}{4}\sum_{q=1}^{Q}\mathbb{E}\|\nabla F(\theta_q)\|^2 \quad \leq & \mathbb{E}[F(\theta_0)]+\frac{T\gamma Q}{2}\left(\frac{3LN\gamma\sigma^2}{(\Gamma^*)^2}+\frac{3L^2\gamma^2TNG}{2\Gamma^*}\right) \\
& +\frac{T\gamma}{2}\left(\frac{3L^2\delta^2N}{2\Gamma^*}\right)\sum_{q=1}^{T}\mathbb{E}\|\theta_q\|^2.
\end{aligned}
\tag{28}
$$

Dividing both sides above by $4/(QT\gamma)$ and choosing $\gamma=1/\sqrt{TQ}$, we have:

$$
\begin{aligned}
\frac{1}{Q}\sum_{q=1}^{Q}\mathbb{E}\|\nabla F(\theta_q)\|^2 \quad \leq & \frac{4\mathbb{E}[F(\theta_0)]}{\sqrt{TQ}}+\frac{6LN\sigma^2}{\sqrt{TQ}(\Gamma^*)^2}+\frac{3L^2NG}{Q\Gamma^*} \\
& +\frac{3L^2\delta^2N}{\Gamma^*}\cdot\frac{1}{Q}\sum_{q=1}^{T}\mathbb{E}\|\theta_q\|^2 \\
=& \frac{G_0}{\sqrt{TQ}}+\frac{V_0}{\sqrt{Q}}+\frac{I_0}{\Gamma^*}\cdot\frac{1}{Q}\sum_{q=1}^{T}\mathbb{E}\|\theta_q\|^2,
\end{aligned}
\tag{29}
$$

where we introduce constants $G_0 = 4\mathbb{E}[F(\theta_0)] + 6LN\sigma^2/(\Gamma^*)^2$, $V_0 = 3L^2NG/\Gamma^*$, and $I_0 = 3L^2\delta^2N$. This completes the proof of Theorem 1.

Finally, for non-IID data distribution, we plug the two upperbounds into Eq.(26) and re-arrange the terms. We follow a similar procedure and choose learning rate $\gamma = 1/\sqrt{TQ}$ and $\gamma \leq 1/(6LT)$. It is straightforward to show that for non-IID data distribution:

$$\frac{1}{Q}\sum_{q=1}^{Q}\mathbb{E}\|\nabla F(\theta_q)\|^2 \leq \frac{H_0}{\sqrt{TQ}} + \frac{V_0}{\sqrt{Q}} + \frac{I_0}{\Gamma^*} \cdot \frac{1}{Q}\sum_{q=1}^{T}\mathbb{E}\|\theta_q\|^2, \tag{30}$$

where $H_1 = 4\mathbb{E}[F(\theta_0)] + 6LK\sigma^2$ is a different constant. This completes the proof of Theorem 2.

## B    Experimental Details

### B.1    Experiment Setup

The code implementation is open sourced and can be found at

Github Link(Link anonymized, see supplementary materials for code and other tools).

In this experimental section we evaluate different pruning techniques from state-of-the-art designs and verify our proposed theory under unifying pruning framework using two datasets.

Unless stated otherwise, the accuracy reported is defined as

$$\frac{1}{n}\sum_i p_i \sum_j \mathrm{Acc}(f_i(x_j^{(i)}, \theta_i \odot m_i), y_j^i))$$

averaged over three random seeds with same random initialized starting $\theta_0$. Some key hyperparameters includes total training rounds $Q = 100$, local training epochs $T = 5$, testing batch size $bs = 128$ and local batch size $bl = 10$. Momentum for SGD is set to 0.5. standard batch normalization is used.

We focus on three points in our experiments: (i) the general coverage of federated learning with heterogeneous models by pruning (ii) the impact of coverage index $\Gamma_{min}$ (iii) the impact of mask error $\delta$.

We examine theoretical results on the following two common image classification datasets: MNIST and CIFAR10, among $N = 100$ workers with IID and non-IID data with participation ratio $c = 0.1$. For IID data, we follow the design of balanced MNIST by previous research, and similarly obtain balanced CIFAR10. For non-IID data, we obtained balanced partition with label distribution skewed, where the number of the samples on each device is up to at most two out of ten possible classifications.

We run all experiments on the small model architecture: an MLP with a single hidden layer for MNIST and a LeNet-5 like network with 2 convolutions layers for CIFAR10. As some large DNN models are proved to have the ability to maintain their performance within a reasonable level of pruning, we use smaller networks to avoid the potential influence from very large networks, as well as other tricks and model characteristics of each framework. Note that the accuracy is NOT directly comparable to models with enormous sizes in some other works.

### B.2    Pruning Techniques

In the paper we select 4 pruning techniques as baselines and we elaborate the details of them. Let $P_m = \frac{\|m\|_0}{|\theta|}$ be the sparsity of mask $m$, e.g.,$P_m = 75\%$ for a model when 25 % of its weights are pruned, and M is the number of the parameters in the model. Then a mask for weights pruning can be defined as:

$$m_i = \begin{cases} 1 & \text{, if } argsort(\theta[i]) < P_m * M \\ 0 & \text{, otherwise} \end{cases}, i \in M \tag{31}$$

Similarly we have the defination for neuron pruning:

$$m_i = \begin{cases} 1 & \text{, if } argsort(\sum \theta_i) < P_m * N \\ 0 & \text{, otherwise} \end{cases}, \theta_i \in \textbf{Neuron } i \tag{32}$$

where N is the total number of neurons in the network, and fixed subnetwork:

$$m_i = \begin{cases} 1 & \text{, if } i < P_m * M \\ 0 & \text{, otherwise} \end{cases}, i \in M \tag{33}$$

where M is the total number of parameters in the network.

Note in adaptive pruning such mask is subject to change after each round of global aggregation. For pruning with pre-trained mask, the mask is generated based on eq(x) for first 3 rounds then fixed for the rest of the training.

An illustration of those pruning techniques can be found in figure.

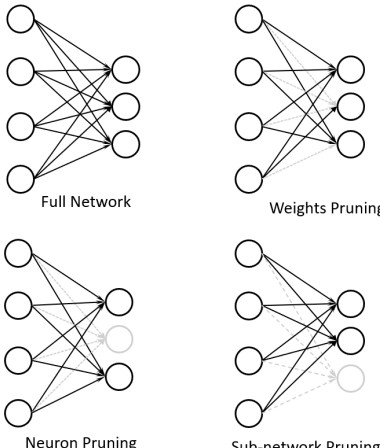

Figure 1: Illustration of pruning techniques used in this paper

## C More Results on MNIST dataset

In this section we present more supplementary experimental results on MNIST dataset. Specifically, we present the training progress in respect of global loss and accuracy for selected pruning techniques. For the final training results we focus on WP, FS and NP as PT is not found competitive without a carefully designed algorithm, however we still keep the training details for PT.

### C.1 Change of Notations

In the main paper we use code name for simplicity of notation and better understanding. Here we present the results with their detailed settings.

For a full model without pruning it can be described as $\mathbb{P}_1(\theta) = \{S_1, S_2, S_3, S_4\}$, where

$$m_i = 1 \text{ if } \theta_i \in \{S_1 \cup S_2 \cup S_3 \cup S_4\} \text{ otherwise } m_i = 0$$

.

Similarly we have another 6 pruning polices as follows:

$$\mathbb{P}_2(\theta) = \{S_1, S_3, S_4\}$$
$$\mathbb{P}_3(\theta) = \{S_1, S_2, S_4\}$$
$$\mathbb{P}_4(\theta) = \{S_1, S_2, S_3\}$$

$$\mathbb{P}_5(\theta) = \{S_2, S_3\}$$
$$\mathbb{P}_6(\theta) = \{S_1, S_3\}$$
$$\mathbb{P}_7(\theta) = \{S_1, S_2\}$$

And we further denote a local client with its pruning policy, as an example, the case "*WP-M1" uses 4 local clients with full models, 2 local clients with pruned models using pruning policy $\mathbb{P}_4$, 2 local clients with pruned models using pruning policy $\mathbb{P}_2$ and 2 local clients with pruned models using pruning policy $\mathbb{P}_3$, then we denote its code name as "1111223344" for simpler notation. Note that we continue to use code name "FedAvg" as a baseline rather than "1111111111". For the rest of the appendix we continue using such notations for denoting its pruning policy settings.

| codename | 1 | 0.75 | 0.5 | PARAs | FLOPs | $\Gamma_{min}$ | %PARA | %FLOPS | IID | Non-IID | |
|---|---|---|---|---|---|---|---|---|---|---|---|
| | | | | | | | | | Accuracy | Global | Local |
| 1111111111 | 10 | | | 159010 | 158800 | 10 | 1.00 | 1.00 | 98.045 | 93.59 | 93.82 |
| 1111114444 | 6 | 4 | | 143330 | 143120 | 6 | 0.90 | 0.90 | 98.18 | 95.15 | 95.49 |
| 1111144447 | 5 | 4 | 1 | 135490 | 135280 | 5 | 0.85 | 0.85 | 97.51 | 89.13 | 89.29 |
| 1111223344 | 4 | 6 | | 135490 | 135280 | 8 | 0.85 | 0.85 | 98.32 | 95.48 | 95.82 |
| 1111234444 | 4 | 6 | | 135490 | 135280 | 6 | 0.85 | 0.85 | 98.39 | 95.45 | 95.96 |
| 1111113477 | 6 | 2 | 2 | 135490 | 135280 | 7 | 0.85 | 0.85 | 96.72 | 91.27 | 91.57 |
| 1111234567 | 4 | 3 | 3 | 123730 | 123520 | 7 | 0.77 | 0.77 | 96.73 | 88.99 | 88.90 |
| 1111444444 | 4 | 6 | | 135490 | 135280 | 4 | 0.85 | 0.85 | 97.85 | 89.13 | 89.29 |
| 1111444477 | 4 | 4 | 2 | 127650 | 127440 | 4 | 0.80 | 0.80 | 96.9 | 93.02 | 93.12 |
| 1111556677 | 4 | | 6 | 111970 | 111760 | 6 | 0.70 | 0.70 | 95.5 | 80.07 | 79.34 |
| 1114556677 | 3 | 1 | 6 | 108050 | 107840 | 5 | 0.67 | 0.67 | 95.80 | 79.30 | 79.75 |
| 1234556677 | 1 | 3 | 6 | 100210 | 100000 | 5 | 0.63 | 0.62 | 95.31 | 81.66 | 81.64 |
| 1455666777 | 1 | 1 | 8 | 92370 | 92160 | 3 | 0.58 | 0.58 | 94.79 | 79.15 | 79.08 |
| 2233445677 | 0 | 6 | 4 | 104130 | 103920 | 5 | 0.65 | 0.65 | 95.95 | 81.27 | 81.17 |
| 1444777777 | 1 | 3 | 6 | 92370 | 92160 | 6 | 0.65 | 0.65 | 95.10 | 72.19 | 71.64 |

Table 1: Results For Weights Pruning on MNIST

| codename | 100% | 75% | 50% | PARAs | FLOPs | $\Gamma_{min}$ | %PARA | %FLOPS | IID | Non-IID | |
|---|---|---|---|---|---|---|---|---|---|---|---|
| | | | | | | | | | Accuracy | Global | Local |
| 1111111111 | 10 | | | 159010 | 158800 | 10 | 1.00 | 1.00 | 98.13 | 95.31 | 95.33 |
| 1111114444 | 6 | 4 | | 143110 | 142920 | 6 | 0.90 | 0.90 | 97.97 | 93.60 | 93.82 |
| 1111144447 | 5 | 4 | 1 | 135160 | 134980 | 5 | 0.85 | 0.85 | 97.39 | 91.92 | 92.18 |
| 1111223344 | 4 | 6 | | 135160 | 134980 | 8 | 0.85 | 0.85 | 97.86 | 91.90 | 92.42 |
| 1111234444 | 4 | 6 | | 135160 | 134980 | 6 | 0.85 | 0.85 | 97.86 | 92.99 | 92.93 |
| 1111234567 | 4 | 3 | 3 | 123235 | 123070 | 7 | 0.77 | 0.77 | 96.64 | 83.82 | 83.61 |
| 1111444444 | 4 | 6 | | 135160 | 134980 | 4 | 0.85 | 0.85 | 97.53 | 91.80 | 92.07 |
| 1111444477 | 4 | 4 | 2 | 127210 | 127040 | 4 | 0.80 | 0.80 | 96.77 | 84.91 | 85.02 |
| 1111556677 | 4 | | 6 | 111310 | 111160 | 6 | 0.70 | 0.70 | 96.57 | 69.11 | 69.63 |
| 1114556677 | 3 | 1 | 6 | 107335 | 107190 | 5 | 0.67 | 0.67 | 95.34 | 77.53 | 77.70 |
| 1234556677 | 1 | 3 | 6 | 99385 | 99250 | 5 | 0.62 | 0.62 | 95.47 | 72.80 | 72.40 |
| 1455666777 | 1 | 1 | 8 | 91435 | 91310 | 3 | 0.57 | 0.57 | 94.41 | 61.96 | 62.49 |
| 2233445677 | 0 | 6 | 4 | 103360 | 103220 | 5 | 0.65 | 0.65 | 96.37 | 60.23 | 61.01 |
| 1444777777 | 1 | 3 | 6 | 99385 | 99250 | 5 | 0.62 | 0.62 | 95.23 | 60.54 | 61.85 |

Table 2: Results For Neuron Pruning on MNIST

| codename | 100% | 75% | 50% | PARAs | FLOPs | $\Gamma_{min}$ | %PARA | %FLOPS | IID | Non-IID | |
|---|---|---|---|---|---|---|---|---|---|---|---|
| | | | | | | | | | Accuracy | Global | Local |
| 1111111111 | 10 | | | 159010 | 158800 | 10 | 1.00 | 1.00 | 97.67 | 94.12 | 94.45 |
| 1111114444 | 6 | 4 | | 143110 | 142920 | 6 | 0.9 | 0.90 | 97.76 | 92.33 | 92.55 |
| 1111144447 | 5 | 4 | 1 | 135160 | 134980 | 6 | 0.85 | 0.85 | 97.34 | 93.79 | 93.92 |
| 1111444444 | 4 | 6 | | 135160 | 134980 | 4 | 0.85 | 0.85 | 97.62 | 92.05 | 92.33 |
| 1111444477 | 4 | 4 | 2 | 127210 | 127040 | 4 | 0.80 | 0.80 | 97.32 | 92.67 | 92.95 |
| 1111444777 | 4 | 3 | 3 | 123235 | 123070 | 4 | 0.77 | 0.77 | 97.35 | 91.34 | 91.73 |
| 1111777777 | 4 | | 6 | 111310 | 111160 | 4 | 0.70 | 0.70 | 97.18 | 93.6 | 93.48 |
| 1114777777 | 3 | 1 | 6 | 107335 | 107190 | 3 | 0.67 | 0.67 | 97.12 | 93.7 | 93.57 |
| 1444777777 | 1 | 3 | 6 | 99385 | 99250 | 1 | 0.62 | 0.62 | 97.01 | 90.74 | 90.57 |
| 1477777777 | 1 | 1 | 8 | 91435 | 91310 | 1 | 0.57 | 0.57 | 96.88 | 90.73 | 90.67 |

Table 3: Results For Fixed Sub-network on MNIST

## C.2 More Results

### C.2.1 Case for IID data

We present the full results of training for IID case in Fig 2 - 5

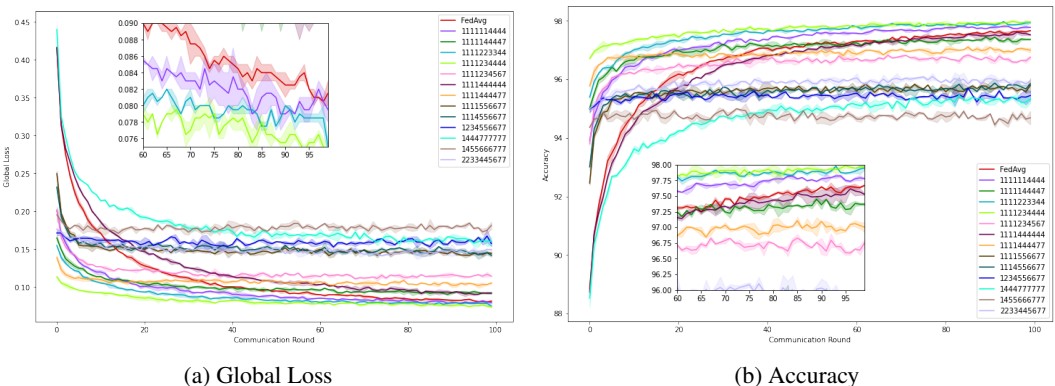

(a) Global Loss                    (b) Accuracy

Figure 2: Results on Weights Pruning on MNIST IID

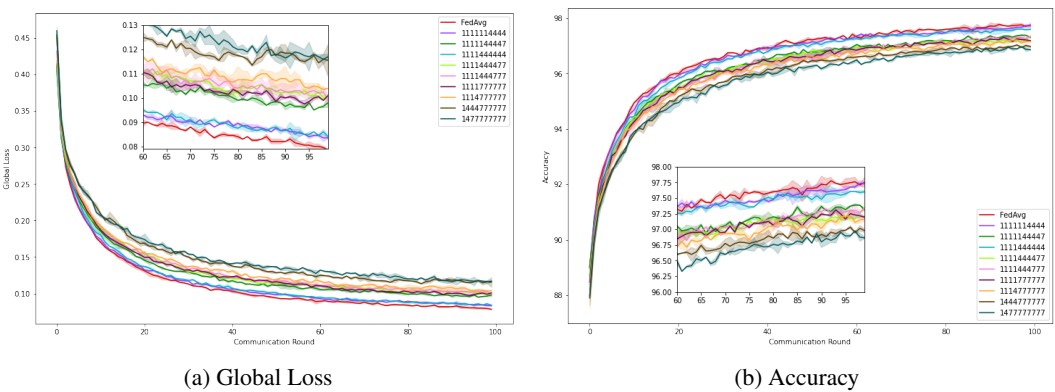

(a) Global Loss                    (b) Accuracy

Figure 3: Results on Fixed Sub-network on MNIST IID

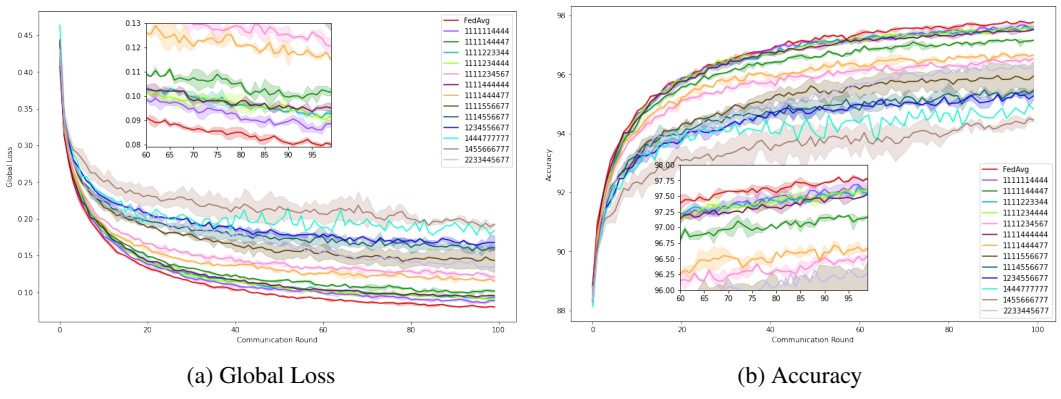

(a) Global Loss                    (b) Accuracy

Figure 4: Results on Neuron Pruning on MNIST IID

### C.2.2 Case for non-IID data

We present the full results of training for non-IID case in Fig 6 - 9

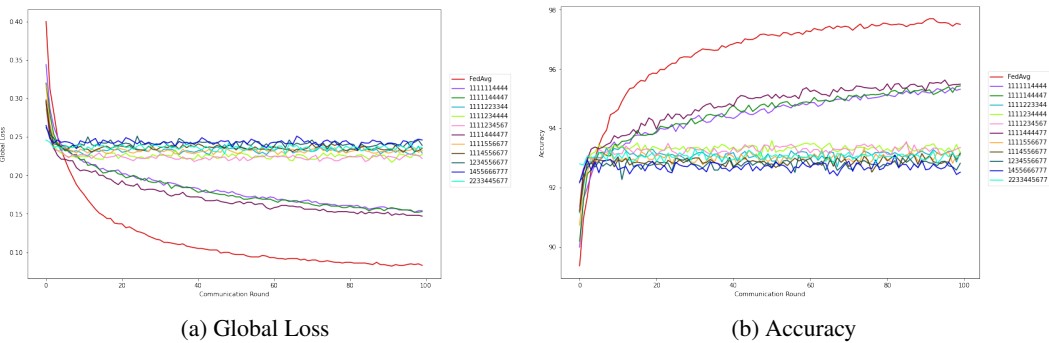

(a) Global Loss

(b) Accuracy

Figure 5: Results on Pruning with pre-trained mask on MNIST IID

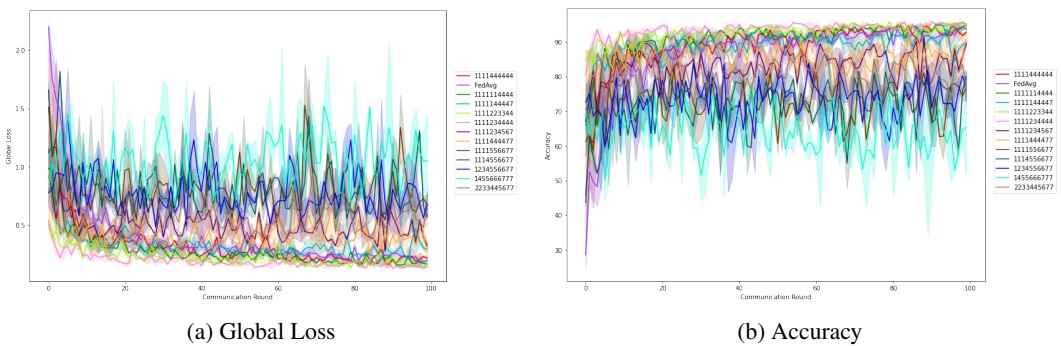

(a) Global Loss

(b) Accuracy

Figure 6: Results on Weights Pruning on MNIST non-IID

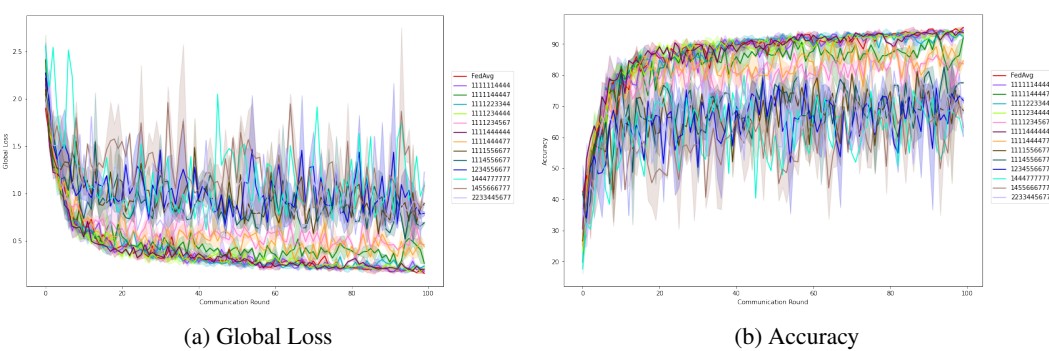

(a) Global Loss

(b) Accuracy

Figure 7: Results on Neuron Pruning on MNIST non-IID

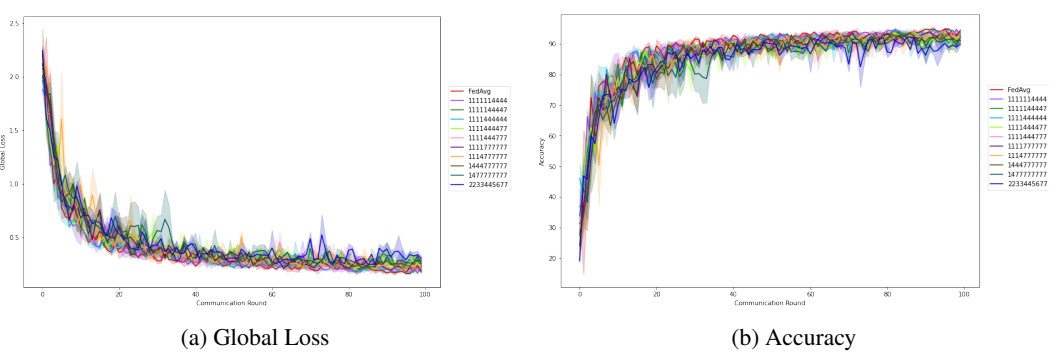

(a) Global Loss

(b) Accuracy

Figure 8: Results on Fixed Sub-network on MNIST non-IID

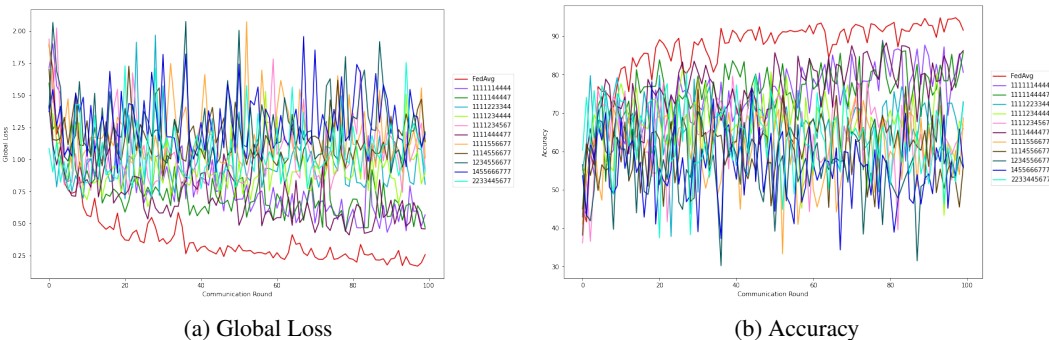

(a) Global Loss

(b) Accuracy

Figure 9: Results on Pruning with pre-trained mask on MNIST non-IID

# D  More Results For CIFAR-10-IID

In this section we present more supplementary experimental results on CIFAR 10 dataset to test the effects of pruning on convolutional layers. Specifically, we present the training progress in respect of global loss and accuracy for selected pruning techniques where we focus on WP and FS.

## D.1  Change of Notations

In the main paper we use code name for simplicity of notation and better understanding. Here we present the results with their detailed settings.

For a full model without pruning it can be described as $\mathbb{P}_1(\theta) = \{S_1, S_2, S_3, S_4\}$, where

$$m_i = 1 \text{ if } \theta_i \in \{S_1 \cup S_2 \cup S_3 \cup S_4\} \text{ otherwise } m_i = 0$$

. As we have demonstrated the effects of pruning MLP layers, on CIFAR10 datasets we focus on the effects of conv2d layers.

We have another 3 pruning polices for conv2d layers as follows:

$$\mathbb{P}_2(\theta) = \{S_1, S_3, S_4\}$$
$$\mathbb{P}_3(\theta) = \{S_1, S_2, S_4\}$$
$$\mathbb{P}_4(\theta) = \{S_1, S_2, S_3\}$$

For WP and PT, when using $\mathbb{P}_2$ the top 75% of kernels will be kept, i.e. for the first conv2d layer, the 5 largest kernels out of total of 6 kernels will be kept, and the 6-th kernel will be pruned. Under all pruning polices MLP layers will be pruned at 75% accordingly. Note under such settings, code name without full model '1', e.g. '2222333444', will not satisfy our necessary condition of convergence.

For FS, we denote $\mathbb{P}_2$ as the similar policy as above but only the first continuous parameters, i.e. for the first conv2d layer, the first 5 kernels out of total of 6 kernels will be kept, and the 6-th kernel will be pruned, together with pruning MLP layers at 75%. We denote $\mathbb{P}_3$ as only pruning conv2d layers and $\mathbb{P}_4$ as only pruning MLP layers. In this case, note that even with same codename for WP and FS, their results are NOT directly comparable.

And we further denote a local client with its pruning policy, as an example, the case "*WP-M1" uses 4 local clients with full models, 2 local clients with pruned models using pruning policy $\mathbb{P}_4$, 2 local clients with pruned models using pruning policy $\mathbb{P}_2$ and 2 local clients with pruned models using pruning policy $\mathbb{P}_3$, then we denote its code name as "1111223344" for simpler notation. Note that we continue to use code name "FedAvg" as a baseline rather than "1111111111". For the rest of the appendix we continue using such notations for denoting its pruning policy settings. For the final training results we focus on WP, FS and NP as PT is not found competitive without a carefully designed algorithm, however we still keep the training details for PT.

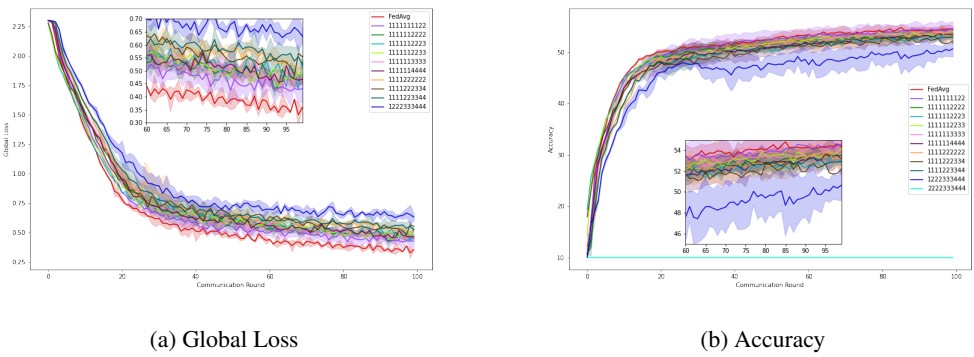

(a) Global Loss  (b) Accuracy

Figure 10: Results on Weights Pruning on CIFAR10 IID

| Codename | PARAs(K) | % | FLOPs(K) | % | Testing Accuracy |
|---|---|---|---|---|---|
| 1111111111 | 512.80 | 1.00 | 653.8 | 1.00 | 53.63 |
| 1111111122 | 482.34 | 0.94 | 619.6 | 0.94 | 53.12 |
| 1111112222 | 451.936 | 0.88 | 587.0 | 0.89 | 52.66 |
| 1111112223 | 451.936 | 0.88 | 587.0 | 0.89 | 52.98 |
| 1111112233 | 451.936 | 0.88 | 587.0 | 0.89 | 54.20 |
| 1111113333 | 451.936 | 0.88 | 587.0 | 0.89 | 52.96 |
| 1111114444 | 451.936 | 0.88 | 587.0 | 0.89 | 51.61 |
| 1111222222 | 421.504 | 0.82 | 553.7 | 0.84 | 51.69 |
| 1111222334 | 421.504 | 0.82 | 553.7 | 0.84 | 52.20 |
| 1111223344 | 421.504 | 0.82 | 553.7 | 0.84 | 52.54 |
| 1222333444 | 375.856 | 0.73 | 503.6 | 0.77 | 49.15 |

Table 4: Results For Weights Pruning on CIFAR 10

| Codename | PARAs(K) | % | FLOPs(K) | % | Testing Accuracy |
|---|---|---|---|---|---|
| 1111111111 | 512.81 | 1.00 | 653.80 | 1.00 | 54.78 |
| 1111111122 | 476.37 | 0.92 | 619.68 | 0.94 | 54.10 |
| 1111112222 | 439.93 | 0.85 | 585.57 | 0.89 | 52.87 |
| 1111113333 | 471.28 | 0.91 | 589.48 | 0.90 | 53.96 |
| 1111113344 | 467.92 | 0.91 | 589.06 | 0.90 | 53.90 |
| 1111114444 | 464.57 | 0.90 | 588.64 | 0.90 | 54.44 |
| 1111222222 | 403.49 | 0.78 | 551.46 | 0.84 | 52.74 |
| 2222333444 | 372.59 | 0.72 | 488.47 | 0.74 | 52.35 |

Table 5: Results For Fixed Sub-network on CIFAR 10

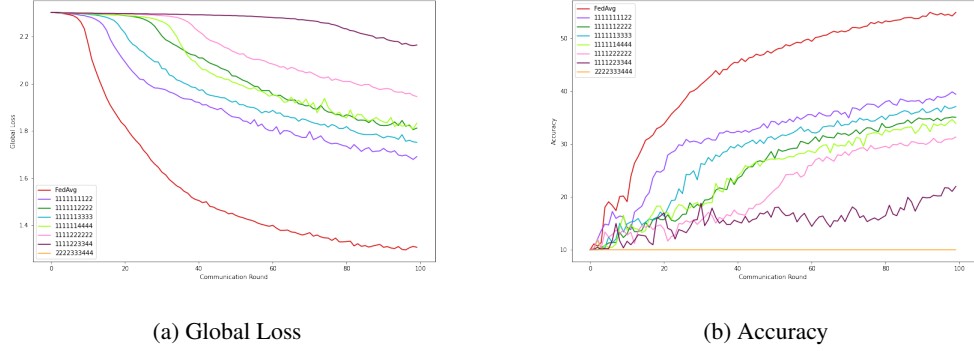

(a) Global Loss  (b) Accuracy

Figure 11: Results on Pruning with pre-trained mask on CIFAR10 IID

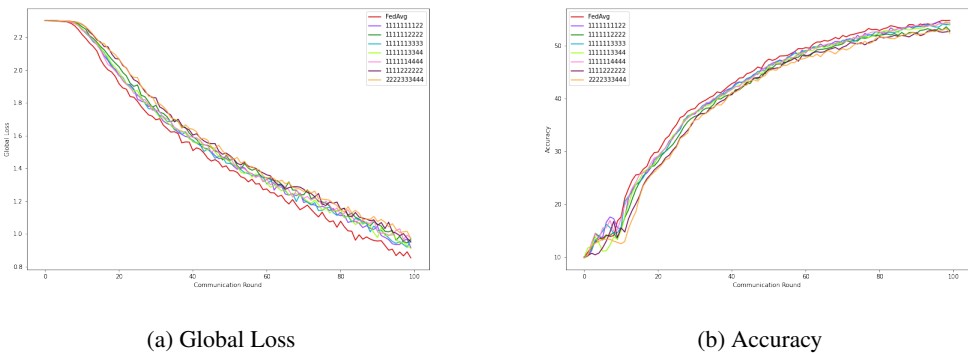

(a) Global Loss

(b) Accuracy

Figure 12: Results on Fixed Sub-network Pruning on CIFAR10 IID

