# OpenReview forum: "Federated Learning with Online Adaptive Heterogeneous Local Models"
_NeurIPS.cc/2022/Workshop/Federated_Learning — FL-NeurIPS 2022 Oral_

### Official Review · Reviewer_pzNM · 2022-10-17
**This paper provides theoretical analysis to the heterogeneous local model training in FL**

The paper theoretical studies the convergence of the recent heterogeneous local model training, where each client only trains a pruned model and this this both computation and communication. The authors prove the convergence of $O(\frac{1}{\sqrt{Q}})$, and also provide empirical studies over various aspects of the heterogeneous local model training.

---

### Official Review · Reviewer_eFuR · 2022-10-17
**The theory probably needs more explanation**

This paper studies model pruning in federated learning. A convergence analysis is shown under the assumption that the pruning induced noise is bounded. An interesting factor \Gamma_{min}, called minimum covering index, is introduced and used to make key observations of various pruning methods. Experiments on several existing pruning methods on MNIST and CIFAR-10 datasets are used to verify the theory.

The insights of theorem 1 is not entirely clear to me. It looks to me the third term on the right hand side will always be there, and can have arbitrary values. Not sure how we can use this to understand the convergence.

Minor issue: there seems to be some typos in the introduction.

---

### Official Review · Reviewer_ysDn · 2022-10-18
**Useful theoretical guarantee for heterogeneous federated ecosystems**

The work's chief contribution is a theoretical guarantee for heterogenous models to converge to a stationary point through federated averaging. This closes a gap in theory left by interesting protocols such as PruneFL and HeteroFL.

Pros:
-Closes an important theoretical gap for federated learning using heterogeneous sub-models of a global model
-Usage of an intuitive parameter coverage index in the presented theorem. The theorem is actually stated quite simply using well-used parameters and constants common to other literature.
-Clear presentation and writing.

Cons:
-A few typos here and there, please proofread the submission.
-Strangely formatted table between lines 211-212. Please make sure it has its own dedicated caption and numbering. It's currently just smashed between blocks of text.

---

### Decision · Program_Chairs · 2022-10-20

Accept (Oral)